# The Antioxidant and Hypolipidemic Effects of *Mesona Chinensis Benth* Extracts

**DOI:** 10.3390/molecules27113423

**Published:** 2022-05-26

**Authors:** Luhua Xiao, Xiaoying Lu, Huilin Yang, Cuiqing Lin, Le Li, Chen Ni, Yuan Fang, Suifen Mo, Ruoting Zhan, Ping Yan

**Affiliations:** 1College of Traditional Chinese Medicine, Guangzhou University of Chinese Medicine, Guangzhou 510006, China; xiaoluhua696@outlook.com (L.X.); luxiaoyinglu@outlook.com (X.L.); hl.yang0708@outlook.com (H.Y.); lincuiqing2022@outlook.com (C.L.); lelililee@outlook.com (L.L.); 020710@gzucm.edu.cn (C.N.); fangyuan131907@icloud.com (Y.F.); suifenmo@outlook.com (S.M.); 2Key Laboratory of Chinese Medicinal Resource from Lingnan (Guangzhou University of Chinese Medicine), Ministry of Education, Guangzhou 510006, China; 3Joint Laboratory of Nation Engineering Research Center for the Pharmaceutics of Traditional Chinese Medicines, Guangzhou 510006, China

**Keywords:** *Mesona Chinensis Benth*, antioxidant, hypolipidemic, component

## Abstract

In this study, the antioxidant and hypolipidemic effects of *Mesona Chinensis Benth* (MCB) extracts were evaluated. Seven fractions (F0, F10, F20, F30, F40, F50 and MTF) were obtained from the MCB ethanol extracts. Compared to the commercial antioxidants (vitamin C), MTF and F30 exhibited higher antioxidant activities in the antiradical activity test and the FRAP assay. The half-inhibition concentration (IC50) for MTF and F30 were 5.323 µg/mL and 5.278 µg/mL, respectively. MTF at 200 µg/mL significantly decreased the accumulation of TG in oleic acid (OA)-induced HepG2 cells and reversed the inhibitory effect of Compound C on AMPK (MTF and F30 significantly increased the glucose utilization of insulin-induced HepG2 cells). In addition, the components of MTF were identified by HPLC-MS, which were caffeic acid, quercetin 3-O-galactoside, isoquercetin, astragalin, rosmarinic acid, aromadendrin-3-O-rutinoside, rosmarinic acid-3-O-glucoside and kaempferol-7-O-glucoside. Through statistical correlations by Simca P software, it was found that the main antioxidant and hypolipidemic components of MCB might be caffeic acid, kaempferol-7-O-glucoside, rosmarinic acid-3-O-glucoside and aromadendrin-3-O-rutinoside, which may play important roles in the AMPK pathway. MTF and F30 in MCB could be potential health products for the treatment of hyperlipidemia.

## 1. Introduction

The *Mesona Chinensis Benth* (MCB) is a widely used antioxidant medicine in China [1]. The Chinese Materia Medica explained that MCB might potentially treat hypertension and hyperglycemia [2,3]. MCB has been widely reported to have antioxidant, hypoglycemic and hyperlipidemic activities [4,5].

In the past two decades, there has been growing interest in novel hyperlipidemic drugs. They could potentially slow the progression of many chronic diseases through antioxidant and hypoglycemic effects [6]. Antioxidants can be hydrogen donors removing free radicals produced in plasma and liver [7]. They can also improve the activities of antioxidant enzymes, including SOD, GSH-Px and CAT [8]. The reduction of free radicals prevents damage to tissues and relieves hyperlipidemia, diabetes, cancer and physical aging [9]. Caffeic acid is associated with antioxidant, hyperglycemic and hyperlipidemic effects [10]. Rosmarinic acid was the most abundant polyphenol in *Thymbra spicata* L., ameliorating lipid accumulation, oxidative stress and inflammation in the NAFLD cellular model [11]. The isoquercetin upregulated antioxidant genes and reduced hyperlipidemia and inflammation [12,13]. In a previous study, the ethanol extract of MCB showed antioxidant effects and reduced the lipid levels of hyperlipidemic rats. The flavonoids, polysaccharides and phenolic acids contained in MCB are natural antioxidants [14]. The tea prepared through MCB extracts improved dyslipidemia via MAPK and AMPK pathways [15]. The consumption of MCB attenuated postprandial glucose levels and improved the antioxidant status induced by a high carbohydrate meal in overweight subjects [16]. The content of caffeic acid in MCB is also high. The caffeic acid promoted the oxidative decomposition of glucose, promoting glycogen synthesis and inhibiting gluconeogenesis [17]. The polysaccharides of MCB regulated lipid transport and metabolism by activating MAPK and AMPK pathways in HepG2 cells [18,19]. Quercetin glucoside and quercetin rhamnoside, which are other chemical components of MCB, showed antioxidant effects [20]. Rosmarinic acid inhibited the production of reactive oxygen species (ROS) in human fibroblast cells induced by ultraviolet A radiation (UVA) [21]. In the liver, isoquercetin promoted the phosphorylation of acetyl-CoA carboxylase and increased the expression of PPARα and farnesoid X receptor. In addition, isoquercetin reduced the plasmatic level of glucose and the translocation of glucose transporter 4 to the skeletal muscle sarcolemma [22,23]. MCB might be an effective antioxidant and hyperlipidemic drug. However, it is unclear which chemical components have an effect on hyperlipidemia.

In this study, the MCB extract was separated by macroporous resin to obtain different components. The components were analyzed and identified by high-performance liquid chromatography (HPLC) and high-performance liquid chromatography–mass spectrometry (HPLC–MS). We evaluated their antioxidant, hypoglycemic and hypolipidemic activities. The antioxidant and hypolipidemic effects were tested through the antiradical activity test, the HepG2 lipid accumulation model and the IR-HepG2 model, respectively. The model constructed by HepG2 is simple, fast and associated with a high success rate. Moreover, the model can simulate the characteristics of hepatocyte lipid accumulation and hepatic steatosis. The relationships between the chemical components and antioxidant and hypolipidemic effects were analyzed by statistical analysis. This study explored the correlation between the chemical components of different fractions from MCB and potential antioxidant and hypolipidemic activities.

## 2. Results

### 2.1. Total Flavonoids and Polysaccharides

In the ethanol extract (EE), the content of total polysaccharides was 28.59%. In the macroporous resin fractionated purification of EE, the total flavonoids were mainly distributed in 10–40% ethanol eluate. The content of total flavonoids in the MCB total flavonoids (MTF) was 78.52%. The content of total polysaccharides in MCB crude polysaccharides (MCP) was 53.82% and in the aqueous extract (AE) was 43.66% (Table 1).

### 2.2. HPLC Analysis

The components of EE, AE, MTF and MCP are shown in Figure 1. The components were identified by comparison with authentic samples of the individual compounds. EE, AE and MTF mainly contained caffeic acid, quercetin 3-O-galactoside, isoquercetin, astragalin, rosmarinic acid, aromadendrin-3-O-rutinoside, rosmarinic acid-3-O-glucoside and kaempferol-7-O-glucoside. After enrichment, the peak areas of various components of MTF increased. MCP contained isoquercetin. F0 contained caffeic acid, quercetin 3-O-galactoside and rosmarinic acid. F10 contained caffeic acid, quercetin 3-O-galactoside, isoquercetin, astragalin and rosmarinic acid. F20 mainly contained quercetin 3-O-galactoside, isoquercetin, astragalin and rosmarinic acid. F30 mainly contained quercetin 3-O-galactoside, isoquercetin, astragalin, rosmarinic acid, aromadendrin-3-O-rutinoside and rosmarinic acid-3-O-glucoside. F40 contained isoquercetin, astragalin, rosmarinic acid, aromadendrin-3-O-rutinoside, rosmarinic acid-3-O-glucoside and kaempferol-7-O-glucoside. F50 contained only astragalin.

### 2.3. HPLC-MS Analysis

The ion chromatogram of caffeic acid, quercetin 3-O-galactoside, isoquercetin, astragalin and rosmarinic acid are shown in Figure 2. Based on fragment ions and previous reports in the literature [24,25,26], Peak 6, Peak 7 and Peak 8 were tentatively deduced as aromadendrin-3-O-rutinoside, rosmarinic acid-3-O-glucoside and kaempferol-7-O-glucoside, respectively (Table 2).

### 2.4. Antioxidant Activities

As shown in Figure 3, the DPPH radical scavenging abilities of different extracts of MCB were determined and compared as follows: MTF > EE > AE > MCP and F30 > F10 > F40 > F20 > F0 > F50 (Figure 3). In addition, the DPPH radical scavenging ability of total flavonoids (IC_50_ = 5.323 µg/mL) and 30% ethanol eluent (IC_50_ = 5.278 µg/mL) was similar to Vitamin C (VC) (IC_50_ = 5.565 µg/mL). The antiradical activity test and the FRAP assay evaluated the antioxidant properties of MCB extracts (Figure 4). The antioxidant activity of AE was lower than EE. The antioxidant activity of MTF improved after enrichment and purification, but not with MCP. In addition, the results of the two antioxidant tests were consistent: MTF > F30 > F10 > F40 > F20. Among the extracts with antioxidant effects, the content of flavonoids was higher, and the content of polysaccharides was relatively low.

### 2.5. The Oleic Acid (OA)-Induced HepG2 Model

The MTT assay was used to determine the toxicity of different MCB extracts in HepG2 cells. Low, moderate and high doses (50, 100 and 200 µg/mL, respectively) were selected within the safe range (Table 3).

The absorbance of EE and MTF at low, moderate and high doses was lower than in the model group (*p* < 0.05). The absorbance of AE at high doses and the absorbance of MCP at moderate and high doses were lower than in the model group (*p* < 0.05). In the macroporous resin fractions, the absorbance of F10 (high doses), F20 (low and high doses), F30 (moderate and high doses) and F40 (high doses) were lower than in the model group (*p* < 0.05) (Figure 5A,B).

The triglyceride (TG) content of EE at moderate and high doses and the TG content of MTF at low and high doses decreased significantly after intervention (*p* < 0.05). The TG content of AE at moderate and high doses and the TG content of MCP at high doses significantly decreased (*p* < 0.05) (Figure 6).

The MTF and simvastatin effectively reduced the lipid accumulation by OA (*p* < 0.05, *p* < 0.01). A similar trend was observed by Compound C intervention (*p* = 0.088, *p* = 0.056). The MTF and simvastatin significantly reduced the TG accumulation by Compound C (*p* < 0.05, *p* < 0.01) (Figure 7).

The results showed that when MTF replaced simvastatin, the same beneficial effect was described with Compound C. The MTF might reverse the inhibitory effect of Compound C on AMPK (Figure 8).

### 2.6. Glucose Intake in IR-HepG2 Cells

The EE group at moderate doses and the MTF group at low, moderate and high doses significantly increased the cell glucose consumption (*p* < 0.05). The cell glucose consumption of the MTF group at high doses was higher than the metformin group (*p* < 0.05). After treatment with the macroporous resin fractionated eluent, the cell glucose consumption was increased (*p* < 0.05). The glucose consumption of the F30 group at different doses was higher than the metformin group (*p* < 0.01) (Figure 9). Comparing the glucose consumption among different macroporous resin fractionated eluents, the effect of F30 was the most significant (*p* < 0.01) (Figure 10).

### 2.7. Statistical Correlations

The PCA and OPLS-DA analysis were used to reveal the main chromatographic peaks of antioxidant and hypolipidemic compounds of MCB. The findings showed significant differences between MTF and other groups (Figure 11). The variable importance projection value (VIP) of the OPLS-DA model was >1.0 [27]. The main chromatographic peaks involved in antioxidant and hypoglycemic activities were as follows: caffeic acid (Peak 1), kaempferol-7-O-glucoside (Peak 8), rosmarinic acid-3-O-glucoside (Peak 7) and aromadendrin-3-O-rutinoside (Peak 6). These components might be the main antioxidant and hypolipidemic components of MCB (Figure 12 and Figure 13).

## 3. Discussion

The extracts of MCB previously showed antioxidant and hypoglycemic activities [3]. In this study, we showed that the antioxidant activity of MTF was significantly higher compared to other fractions (including polysaccharides) in MCB (Figure 3 and Figure 4). The MTF showed higher hydroxyl free radical scavenging activity than vitamin C (Figure 2). Of note, MTF and simvastatin showed similar effects in reducing lipid accumulation (Figure 8). MCB attenuated postprandial glucose levels by improving the antioxidant status [16]. MTF was also able to increase the consumption of glucose (Figure 8). The description of effective antioxidant and hypolipidemic components of MCB will support the development of new medications.

The chemical constituents of MCB were explored by HPLC and HPLC–MS. The relationship among chemical components, antioxidant and hypolipidemic activities was tested by statistical correlation. The main antioxidant and hypolipidemic components of MCB were caffeic acid, kaempferol-7-O-glucoside, rosmarinic acid-3-O-glucoside and aromadendrin-3-O-rutinoside. The results of AMPK inhibition showed that MTF activated the AMPK pathway. Caffeic acid in MTF has been reported to target the AMPK signaling pathway and regulate oxidative metabolism and glycolysis [28]. Kaempferol-7-O-glucoside modulated lipid and glucose metabolism by upregulating adiponectin and AMPK in obese mice [29]. In addition, kaempferol-7-O-glucoside prevented the inactivation of AKT and AMPK, playing an important role in glycometabolism [30]. Rosmarinic acid-3-O-glucoside displayed significant antioxidant activity due to the inhibition of the AMPK/mTOR signaling pathway [31]. Aromadendrin-3-O-rutinoside improved insulin resistance via PI3K- and AMPK-dependent pathways, thus being a potential candidate for the management of type 2 diabetes mellitus [32]. These studies revealed that caffeic acid, kaempferol-7-O-glucoside, rosmarinic acid-3-O-glucoside and aromadendrin-3-O-rutinoside played antioxidant and hypolipidemic roles through the AMPK pathway.

It is necessary to confirm these hypotheses. The hypolipidemic mechanism of caffeic acid, kaempferol-7-O-glucoside, rosmarinic acid-3-O-glucoside and aromadendrin-3-O-rutinoside needs to be verified in vivo. In future studies, we will use pure compounds to verify their antioxidant and hypolipidemic activities.

## 4. Materials and Methods

### 4.1. Chemicals and Reagents

2,2-diphenyl-1-picrylhydrazyl (DPPH) and dimethyl sulfoxide (DMSO) were purchased from Sigma-Aldrich (USA). Dulbecco’s modified eagle medium (DMEM), Phosphate buffered saline (PBS) and fetal bovine serum (FBS) were purchased from GIBCO (USA). Reference standards of caffeic acid (purity ≥ 99%), astragalin (purity ≥ 98%) and isoquercetin (purity ≥ 98%) were purchased from Shanghai Winherb Medical Technology Co., Ltd. Rutin, D-glucose anhydrous, quercetin 3-O-galactoside (purity ≥ 98.2%) and rosmarinic acid (purity ≥ 98.8%) were purchased from the National Institutes for Food and Drug Control. Oil Red O was purchased from Phygene Life Sciences Company. Simvastatin was purchased from Meilun Biotechnology Co., Ltd. Oleic acid was purchased from TCI (Japan). The TG assay kit was purchased from the Nanjing Jiancheng Bioengineering Institute.

### 4.2. Extraction Procedure

MCB was provided by Guangdong NanLing Pharmaceutical Co., Ltd. (Guangdong, China). MCB was crushed into fine powders, heated and refluxed with 50% ethanol (*v*/*v*) for 1.5 h. Then, the material was filtered and freeze-dried to obtain the EE. The powders of MCB underwent ultrasonic extraction with distilled water (*v*/*v*) for 0.75 h. Then, it was filtered and freeze-dried to obtain the AE.

An appropriate amount of AE was subsequently dissolved in distilled water with the addition of ethanol. The extract was allowed to stand at 4 °C overnight, then it was centrifuged and the precipitate was washed with ethanol, ethyl acetate and acetone 3 times. The precipitate was later dissolved with an appropriate amount of pure water and the filtrate was freeze-dried to obtain the MCP.

A total of 5 g of X-5 macroporous resin was chosen as purification column material. An appropriate amount of EE was dissolved in distilled water, with a total flavonoid content of 6 mg/mL. The pH was adjusted to 3.0. The loading amount of the EE solution was 35 mL, with a 1.2 mL/min flow rate. The washing solution was three times greater than the column volume of water. Approximately 40 mL of 60% ethanol was used as eluent, with a 1.5 mL/min flow rate. The material was then freeze-dried to obtain the MTF.

The components of EE were separated and enriched in the X-5 macroporous resin column. Then, they were washed and eluted with 10%, 20%, 30%, 40% and 50% volume fractions of ethanol solution. The eluent of each concentration was concentrated under reduced pressure and freeze-dried to obtain F0, F10, F20, F30, F40 and F50.

### 4.3. Chemical Composition Analysis

#### 4.3.1. Total Flavonoids and Polysaccharides

The amount of rutin was dissolved in methanol to obtain a reference solution of 0.2 mg/mL. The D-glucose anhydrous was dissolved in distilled water to obtain a reference solution of 0.1 mg/mL.

The total flavonoids were determined by sodium nitrite-aluminum nitrate colorimetry. A certain amount of MCB extracts was placed in a 25 mL volumetric flask. Approximately 1 mL of 5% sodium nitrite solution was added and mixed for 6 min. Then, 1 mL of 10% aluminum nitrate solution was added and mixed for 6 min. A total of 10 mL of sodium hydroxide solution was later added, along with distilled water, and mixed for 15 min. The absorbance value was measured at 510 nm. The rutin was used for the standard calibration curve.

The total polysaccharides were determined by the anthrone-sulfuric acid method. The MCB extracts were placed in a test tube. Around 3 mL of 0.2% anthrone-sulfuric acid solution was added on ice and mixed. Then, the material was incubated at 80 °C for 20 min. After heating, a cooling step treatment before reading absorbance at 620 nm was performed. The D-glucose anhydrous was used for the standard calibration curve.

#### 4.3.2. HPLC Analysis

The chromatographic separation was performed using the Ultimate 3000 HPLC system (Thermo Corporation, Waltham, MA, USA) equipped with a diode array detector. Chromatographic column: ZORBZX Eclipse XDB-C18 (4.6 × 250 mm, 5 µm). The mobile phase consisted of acetonitrile (A) and 0.2% aqueous formic acid (B) in a gradient elution mode as follows: 0–15 min, 5 A: 95 B (*v*/*v*) to 19 A: 81 B (*v*/*v*); 15–35 min, 19 A: 81 B (*v*/*v*) to 21 A: 79 B (*v*/*v*); 35–40 min, 21 A: 79 B (*v*/*v*) to 28 A: 72 B (*v*/*v*); 40–55 min, 28 A: 72 B (*v*/*v*) to 35 A: 65 B (*v*/*v*). The flow rate was 1.0 mL/min, with a sample injection volume of 10 μL. The detection wavelengths were set at 320 nm.

The amount of caffeic acid, quercetin 3-O-galactoside, isoquercetin, rosmarinic acid and astragalin were appropriately dissolved in methanol to prepare a reference solution of 0.1 mg/mL. The MCB extracts (0.05 g) were dissolved in corresponding solvents, then filtered through 0.22 μm microporous membranes.

#### 4.3.3. HPLC-MS Analysis

The chromatographic separation was performed using the Ultimate 3000 HPLC system (Thermo Corporation, USA) equipped with a Thermo Scientific TSQ Quantiva MS (Thermo Co., Denver, CA, USA) and an electrospray ion source. Chromatographic column: ZORBZX Eclipse XDB-C18 (4.6 × 250 mm, 5 µm). Chromatographic separation conditions have been previously shown. Data were acquired using the Q1MS scan mode. Samples were analyzed under the positive and negative ionization modes. Mass parameters of the electron spray ionization ion source were set as follows: capillary temperature, 320 °C; heater temperature, 300 °C; sheath gas flow rate, 35 arb; auxiliary gas flow rate, 10 arb; scan range, 50–800 m/z.

### 4.4. Antioxidant Activity

#### 4.4.1. Antiradical Activity

The antiradical activity of MCB was determined via the DPPH assay [33]. Around 100 μL of each extract solution at different concentrations (5, 10, 50, 100 and 200 μg/mL) was mixed with 100 μL of DPPH (0.2 mol/mL). The ascorbic acid was used as a positive control. The absorbance was recorded at 517 nm and converted to the radical scavenging activity using the following equation:$$\mathrm{Scavenging}\;\mathrm{activity}\;(\%)=\frac{1-(A_{sample}-A_{blank})}{A_{control}}\;\times \;100\%$$

Each sample was run in triplicate and the results were averaged.

#### 4.4.2. FRAP Assay

The working reagent was prepared by mixing acetate buffer (300 mM), TPTZ (10 mM) and FeCl_3_·6H_2_O (20 mM) at a ratio of 10:1:1 (*v*/*v*/*v*). A total of 20 μL of an MCB extract solution (200 μg/mL) was prepared in 96-well plates with the working reagent (180 μL). After 20 min of incubation, the absorbance was measured at 740 nm. A series of Fe_2_SO_4_ solutions were used to prepare the calibration curve (concentration range: 0–280 µg/mL). The standard curve was drawn as y = 0.0018x + 0.0145, R^2^ = 0.9981.

### 4.5. HepG2 Cell Lipid Accumulation

#### 4.5.1. Cell Culture

The human hepatoblastoma (HepG2) cell line was purchased from the Shanghai Institutes for Biological Sciences. The medium was composed of DMEM, 10% fetal bovine serum, 100 μg/mL streptomycin and 100 μg/mL penicillin in a humidified atmosphere of 5% CO_2_ at 37 °C. A total of 5 × 10^5^ cells/well were cultured using the above medium and inoculated in 24-well plates. When the cells reached 70% confluence, the MCB extracts (EE, AE, MCP, MTF, F0, F10, F20, F30, F40 and F50) were added at various concentrations for different time intervals. An AMPK inhibitor was prepared in DMSO as a 10 mM stock solution.

#### 4.5.2. MTT Cytotoxicity Assay

Each extract was stimulated by the DMEM medium to obtain a sample solution of 4 mg/mL. The obtained cells (5 × 10^3^/well) were seeded in 96-well plates and treated with different concentrations of MCB extracts for 24 h. After 4 h of incubation with MTT dye, blue formazan crystals were dissolved in DMSO, and the absorbance was recorded at 570 nm.

#### 4.5.3. Lipid Accumulation and TG Content

Cells (5 × 10^5^/well) were incubated with OA (0.2 mM) and MCB extracts (50 μg/mL, 100 μg/mL and 200 μg/mL) for 24 h. The culture media were removed, and cells were fixed with paraformaldehyde (PFA) (4%), then washed three times with PBS. After 30 min of incubation with Oil Red O, cells were washed three times with distilled water and observed by microscopy. Later, isopropanol was added, followed by 30 min of incubation. The absorbance was measured at 510 nm. The TG content was measured according to the instruction kit.

#### 4.5.4. The AMPK Inhibitor (Compound C)

Cells (5 × 10^5^/well) were seeded in 96-well plates and divided into the following groups: ① control group; ② OA (0.2 mmol/L) group; ③ OA (0.2 mmol/L) + simvastatin (25 µM) group; ④ OA (0.2 mmol/L) + MTF (200 µg/mL) group; ⑤ Compound C (10 µM) group; ⑥ simvastatin (25 µM) + Compound C (10 µM) group; ⑦ MTF (200 µg/mL) + Compound C (10 µM) group. After a 24 h intervention, culture media were aspirated, and cells were washed three times with PBS. Around 200 µL of cell lysate were added for 20 min. Then, the cells were carefully removed from the surface with a cell scraper and collected in a 1.5 ml EP tube placed on ice. The lipid accumulation was later determined using the method described in 4.5.3.

### 4.6. Insulin Resistance of HepG2 Cell

Cells (5 × 10^5^/well) were seeded in 96-well plates until cell confluence. Then, the culture medium was replaced with DMEM without FBS. After starvation for 12 h, several concentrations of MCB extracts (50, 100, and 200 μg/mL, respectively) were investigated for 24 h with insulin (10 μL). Metformin was used as a positive control. The supernatant culture medium was collected, and the glucose concentration was determined.

### 4.7. Spectral Correlations

The peak areas of EE, AE, MCP, F0, F10, F20, F30, F40 and F50 were standardized. The relationship between peak areas with antioxidant and hypolipidemic activities was analyzed. The score diagram was obtained by Principal Component Analysis (PCA) and Orthogonal Partial Least Squares (OPLS) using Simca P 14.0 software.

## 5. Conclusions

In the present study, MCB extracts and seven fractions were evaluated chemically and pharmacologically. The chemical composition analysis found that MCB mainly contained caffeic acid, quercetin 3-O-galactoside, isoquercetin, astragalin, rosmarinic acid, aromadendrin-3-O-rutinoside, rosmarinic acid-3-O-glucoside and kaempferol-7-O-glucoside. The peak areas of various components of MTF increased after enrichment. MTF showed better antioxidant and hypolipidemic effects than other fractions and significantly increased cell glucose consumption, which may be related to the activation of the AMPK pathway. Through the spectral correlation analysis of chemical components and antioxidant and hypolipidemic activities, caffeic acid, kaempferol-7-O-glucoside, rosmarinic acid-3-O-glucoside and aromadendrin-3-O-rutinoside displayed relevant antioxidant and hypolipidemic activities. The MCB has great potential for application in the treatment of hyperlipidemia.

## Figures and Tables

**Figure 1 molecules-27-03423-f001:**
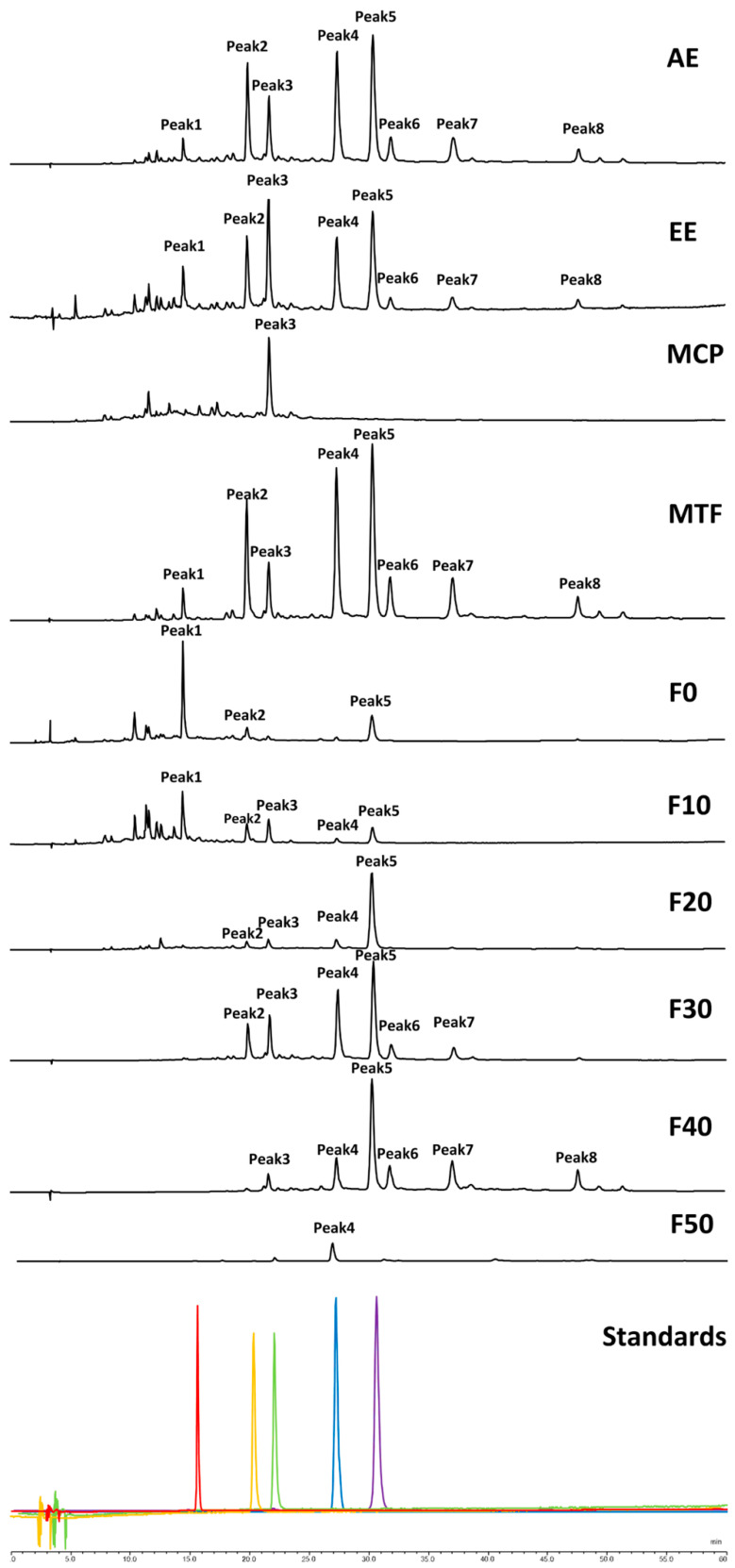
Components of MCB extracts and seven fractions investigated using HPLC. The used wavelength was 320 nm, peaks 1–6 were identified by comparison with authentic standard samples. Peak 1: caffeic acid, Peak 2: quercetin 3-O-galactoside, Peak 3: isoquercetin, Peak 4: astragalin, Peak 5: rosmarinic acid, Peak 6: aromadendrin-3-O-rutinoside, Peak 7: rosmarinic acid-3-O-glucoside, Peak 8: kaempferol-7-O-glucoside.

**Figure 2 molecules-27-03423-f002:**
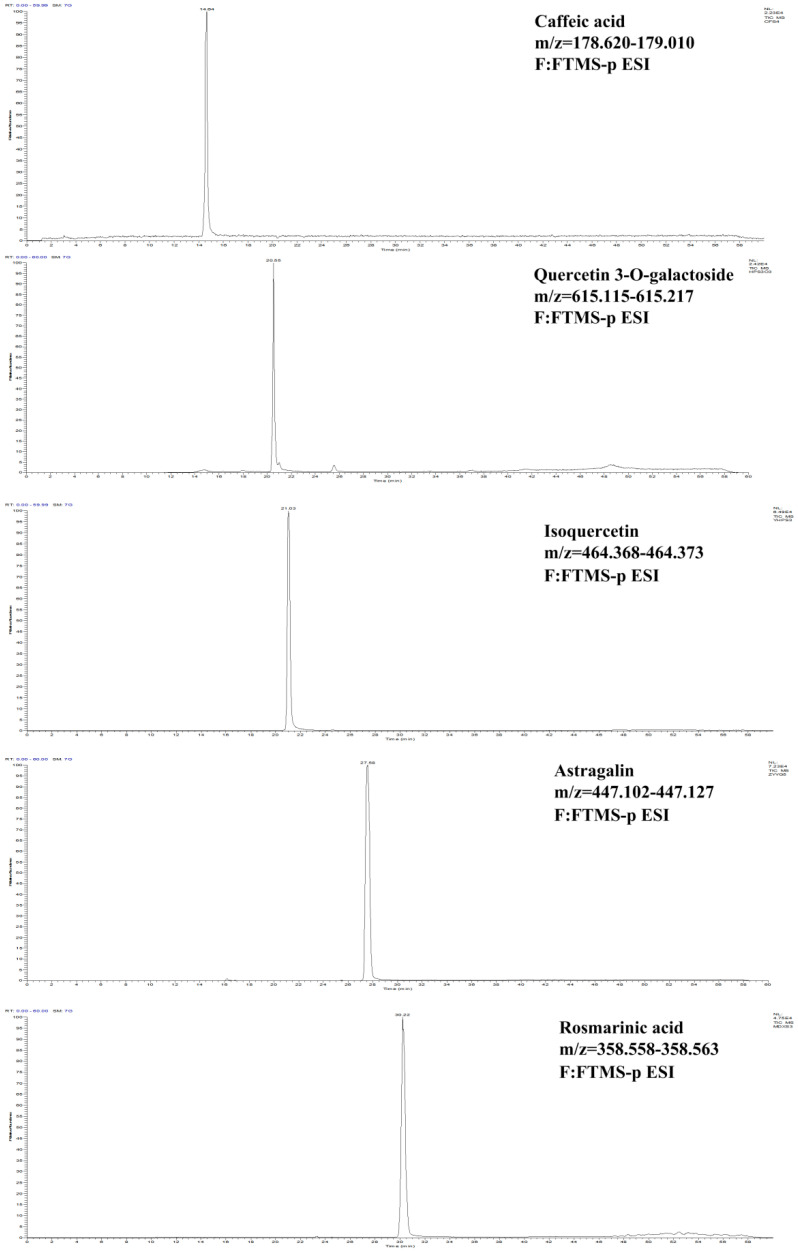
Extract ion chromatogram of standards.

**Figure 3 molecules-27-03423-f003:**
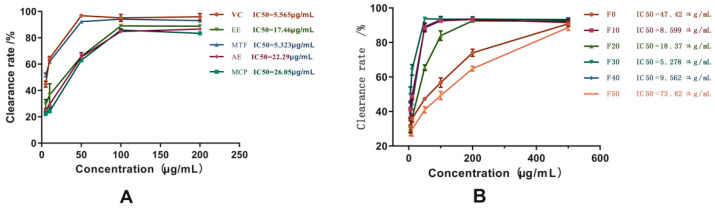
DPPH radical scavenging test results. (**A**) DPPH free radical scavenging ability of alcohol extracts and aqueous extracts before and after purification. (**B**) DPPH radical scavenging ability of the fractional eluent. Data are expressed as mean ± standard deviation (*n* = 5).

**Figure 4 molecules-27-03423-f004:**
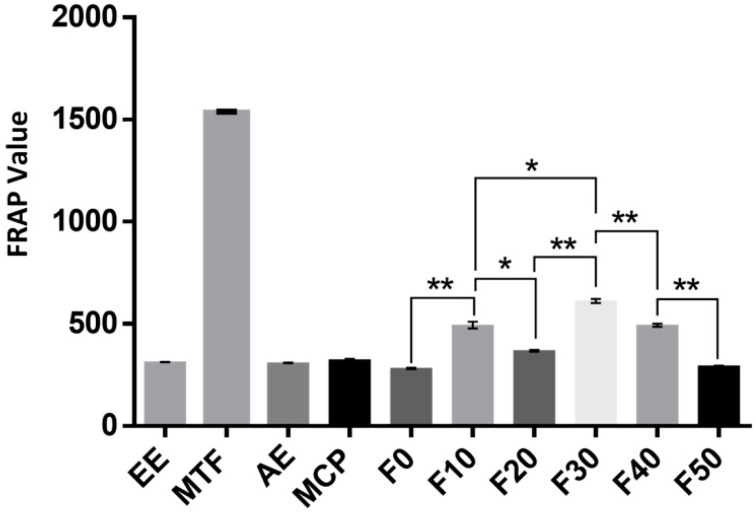
Results of the FRAP method. The in vitro antioxidant activity of MCB is expressed by FRAP values, followed by F30, F10, F40 and F20. EE: alcohol extract, AE: aqueous extract, MTF: total flavonoids, MCP: crude polysaccharides. “*” and “**” mean *p* < 0.01 and *p* < 0.05, respectively. Data are expressed as mean ± standard deviation (*n* = 5).

**Figure 5 molecules-27-03423-f005:**
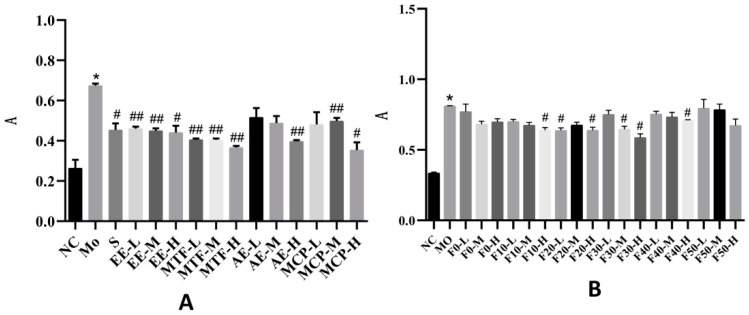
Effects of MCB extracts on total lipid accumulation in HepG2 cells. (**A**) the absorbance of MCB extracts on total lipid accumulation. (**B**) the absorbance of the fractions on total lipid accumulation. NC: control group, Mo: model group, S: drug group, 50: low dose group, M: moderate dose group, H: high dose group, EE: ethanol extract, AE: aqueous extract, MTF: total flavonoids, MCP: crude polysaccharides, macroporous resin fractional eluent: F0, F10, F20, F30, F40 and F50. “*” indicates *p* < 0.05 compared with the NC group. “#” and “##” respectively indicate *p* < 0.05 and *p* < 0.01 compared with the Mo group.

**Figure 6 molecules-27-03423-f006:**
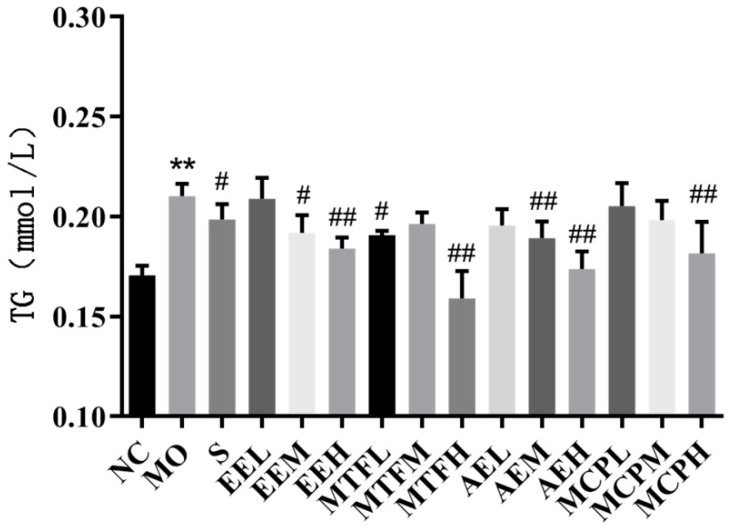
Effects of MCB extracts on TG level in HepG2 cells. NC: control group, Mo: model group, S: drug group; 50: low dose group, M: moderate dose group, H: high dose group, EE: ethanol extract, AE: aqueous extract, MTF: total flavonoids, MCP: crude polysaccharides. “**” indicates *p* < 0.01 compared with the NC group. “#” and “##” respectively indicate *p* < 0.05 and *p* < 0.01 compared with the Mo group.

**Figure 7 molecules-27-03423-f007:**
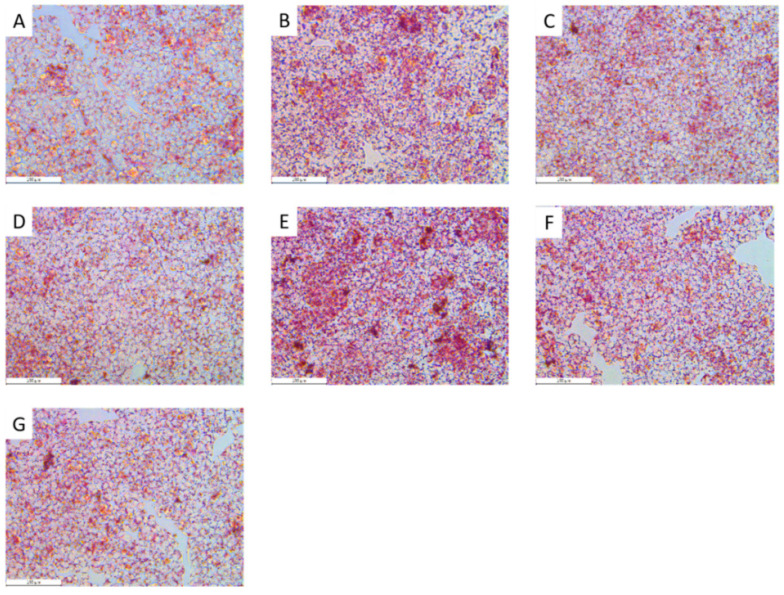
Effects of total flavonoids of MCB on fat accumulation in HepG2 cells (200×). (**A**) control group, (**B**) OA group, (**C**) OA + simvastatin (25 µM) group, (**D**) OA + MTF (200 µg/mL) group, (**E**) Compound C (10 µM) group, (**F**) simvastatin (25 µM) + Compound C (10 µM) group, (**G**) MTF (200 µg/mL) + Compound C (10 µM) group.

**Figure 8 molecules-27-03423-f008:**
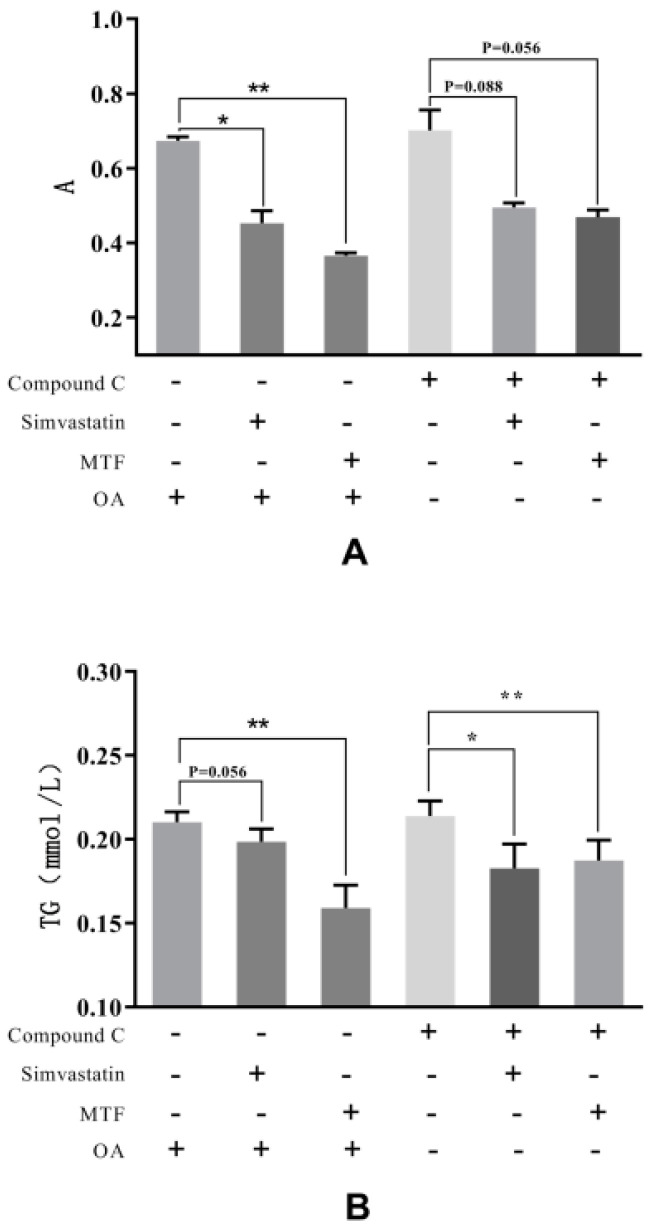
Effects of total flavonoids (MTF) of MCB on lipid accumulation in HepG2 cells. (**A**) the absorbance of lipid accumulation was measured by oil red O staining. (**B**) changes of TG content. “*” indicates *p* < 0.05 and “**” indicates *p* < 0.01.

**Figure 9 molecules-27-03423-f009:**
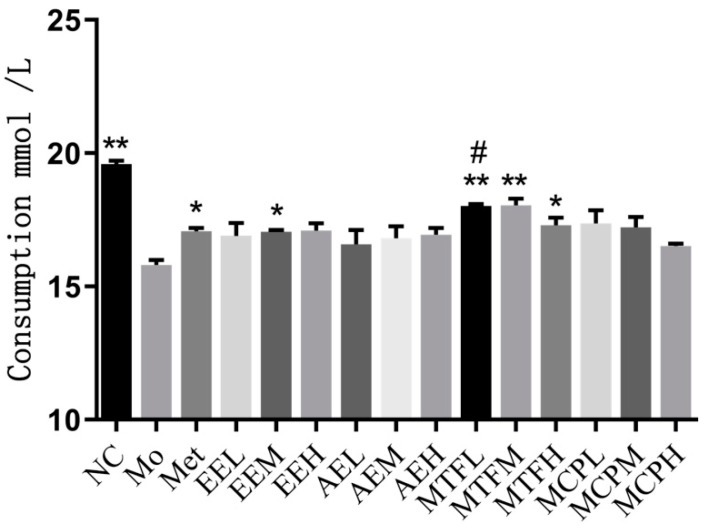
Effects of MCB extracts on glucose intake in IR-HepG2 cells. NC: control group, Mo: model group, Met: drug group, EE: ethanol extract, AE: aqueous extract, MTF: total flavonoids, MCP: crude polysaccharides. “*” and “**” represent *p* < 0.05 and *p* < 0.01 compared with the Mo group, respectively. “#” indicates *p* < 0.05 compared with the Met group.

**Figure 10 molecules-27-03423-f010:**
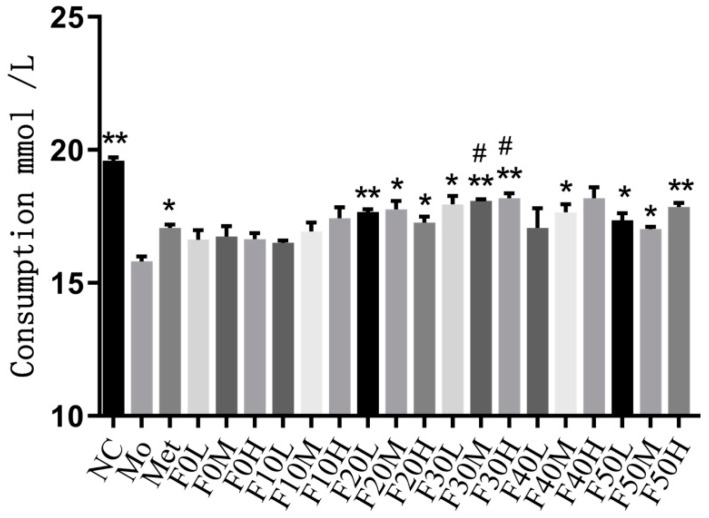
Effects of macroporous resin fractionated eluates on glucose intake in IR-HepG2 cells. NC: control group, Mo: model group, Met: drug group. “*” and “**” represent *p* < 0.05 and *p* < 0.01 compared with the Mo group, respectively. “#” indicates *p* < 0.05 compared with the Met group.

**Figure 11 molecules-27-03423-f011:**
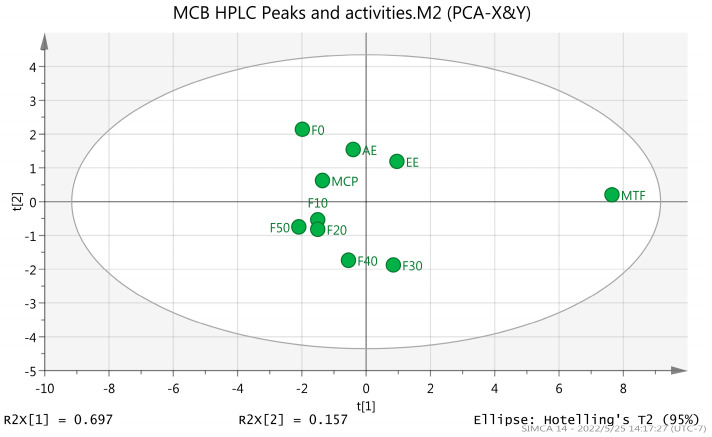
Scores of MCB extracts and fractions (PCA). EE: ethanol extract, AE: aqueous extract, MTF: total flavonoids, MCP: crude polysaccharides, macroporous resin fractional eluent: F0, F10, F20, F30, F40 and F50.

**Figure 12 molecules-27-03423-f012:**
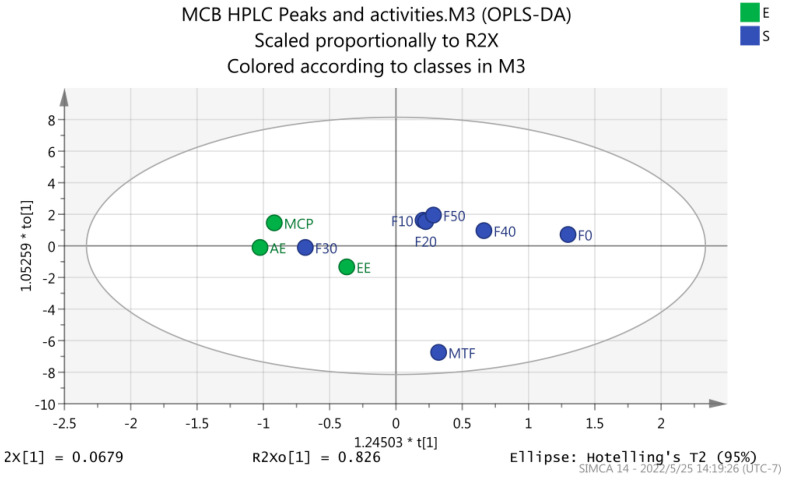
Scores of MCB extracts and fractions (OPLS−DA). EE: ethanol extract, AE: aqueous extract, MTF: total flavonoids, MCP: crude polysaccharides, macroporous resin fractional eluent: F0, F10, F20, F30, F40 and F50.

**Figure 13 molecules-27-03423-f013:**
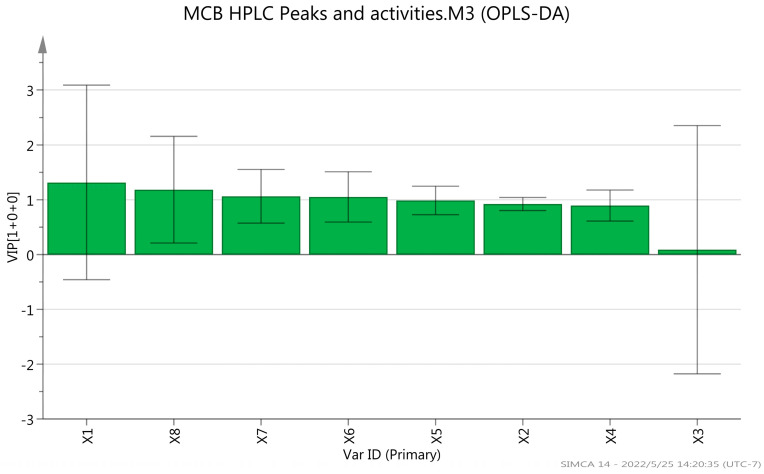
VIP of MCB extracts and fractions. X1 (Peak 1): caffeic acid, X2 (Peak 2): quercetin 3-O-galactoside, X3 (Peak 3): isoquercetin, X4 (Peak 4): astragalin, X5 (Peak 5): rosmarinic acid, X6 (Peak 6): aromadendrin-3-O-rutinoside, X7 (Peak 7): rosmarinic acid-3-O-glucoside, X8 (Peak 8): kaempferol-7-O-glucoside.

**Table 1 molecules-27-03423-t001:** Determination of total flavonoids and polysaccharides.

Sample Name	Total Flavonoids (%)	Total Polysaccharides (%)
F0	5.74	35.27
F10	31.73	12.14
F20	30.73	6.69
F30	65.27	9.32
F40	29.53	9.79
F50	16.95	4.99
AE	17.71	43.66
MCP	16.62	53.82
EE	24.30	28.59
MTF	78.52	18.83

**Table 2 molecules-27-03423-t002:** HPLC-MS analysis of MTF.

NO.	tR (Min)	Observed [M-H]-(m/z)	Formula	Fragment Ions	Compound	Ref.
Peak 1	14.647	178.62	C_9_H_8_O_4_	178.62, 151.01, 134.70, 112.89	Caffeic acid	Standard compound
Peak 2	20.04	615.20	C_28_H_24_O_16_	615.20, 460.13, 391.10, 296.82, 182.90	Quercetin 3-O-galactoside	Standard compound
Peak 3	21.86	301.06	C_15_H_10_O_7_	301.06, 273.01, 150.97, 122.46	Isoquercetin	Standard compound
Peak 4	27.56	447.10	C_21_H_20_O_11_	447.10, 280.52, 242.83, 92.95	Astragalin	Standard compound
Peak 5	30.553	358.56	C_18_H_16_O_8_	358.56, 196.65, 162.79, 137.00	Rosmarinic acid	Standard compound
Peak 6	26.27	592.86	C_27_H_30_O_15_	592.86, 446.58, 326.59	Aromadendrin-3-O-rutinoside	[24]
Peak 7	32.057	520.87	C_24_H_26_O_13_	520.87, 358.78, 196.81, 160.60,135.17	Rosmarinic acid-3-O-glucoside	[25]
Peak 8	37.28	447.10	C_21_H_20_O_11_	447.10, 264.59, 242.83, 150.90	Kaempferol-7-O-glucoside	[26]

**Table 3 molecules-27-03423-t003:** Cytotoxicity of MCB extracts.

Sample	HepG2 Cell Viability IC_50_ (µg/mL)
EE	731.1
AE	401.4
MCP	-
MTF	-
F0	-
F10	630.4
F20	691.0
F40	590.8
F50	502.5

## Data Availability

Not applicable.

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
