# Peer review of "The Antioxidant and Hypolipidemic Effects of Mesona Chinensis Benth Extracts"

_molecules, 2022, doi:10.3390/molecules27113423_

Round 1

Reviewer 1 Report

The manuscript from Xiao et al describes the fractionation of a traditional Chinese medicine and chemical characterization of the composition of those fractions. The antioxidant activity and affect on glucose uptake of the different fractions is reported.

I would have liked to see a comparison of the biological activities of the fractions with the pure compounds that were identified in the fractions. This would have allowed the authors to better attribute activity to specific molecules. The authors themselves state that this work is necessary. Including it here would make a much stronger paper.

A few minor points that need correction are listed below.

line 80 define abbreviations EE, AE, MTF and MCP and VC

Section 2.1 The total flavonoids and polysaccharides reported in this paragraph differ from those reported in Table 1. Which is correct?

Section 2.2 Should mention here components were identified by comparison with authentic samples of the individual compounds.

Figure 1 What wavelength was used to produce these traces? Include in caption that peaks 1-6 were identified by comparison with authentic standard samples.

line 336, 339  subscripts needed in molecular formulas

Author Response

Dear reviewer:

   Thank you very much for your professional suggestions. Comparison of biological activities between fractions and pure compounds is ongoing. At present, we have analyzed 8 pure compounds in Mesona Chinensis Benth. The compounds that can be purchased are caffeic acid, quercetin 3-O-galactoside, isoquercetin, astragalin, rosmarinic acid, aromadendrin-3-O-rutinoside, rosmarinic acid-3-O-glucoside. The biological activities of pure compounds is being studied, and separating and identifying pure compounds that cannot be purchased. And the language of the article has been revised by a professional company.

Reply to your first suggestion: On line 79, we define abbreviations EE, AE, MTF, MCP and VC (line 119).

Reply to your second suggestion: Section 2.1, the total flavonoids and polysaccharides have been reorganized in this paragraph, MTF is the total flavonoids fraction from MCB, and MCP is the total polysaccharides fraction from MCB (line 79).

Reply to your third suggestion: Section 2.2 Added "the composition is determined by comparing with the real sample of a single compound." (line 87).

Reply to your fourth suggestion: The wavelength was used to produce these traces at 320 nm (line 100). In caption, peaks 1-6 were identified by comparison with authentic standard samples (line 100). On line 332 and 334,  the molecular formulas has been subscripted.

Kind regards

Luhua Xiao

Reviewer 2 Report

The manuscript entitled “The antioxidant and hypolipidemic effects of Mesona Chinensis Benth extracts” by Luhua Xiao et al. describes the study of extracts from Mesona Chinensis Benth herbal plant. Authors had prepared Mesona Chinensis Benth ethanol extract and separated it on seven fractions. Components in fractions were identified with high-performance liquid chromatography-mass spectrometry. For each fraction, pharmacological activities were evaluated. Authors found that the MTF fraction is the most promising and shows interesting antioxidant and hypolipidemic effects. Overall, the manuscript can be accepted for publication after minor revision. One remark about Conclusion. The Conclusion is too laconic and must be rewritten.

Author Response

Dear reviewer:

   Thank you very much for your comments and suggestions. The conclusion has been rewritten, the details on line 382.

Kind regards

Ping Yan

Reviewer 3 Report

In the presented article "The antioxidant and hypolipidemic effects of Mesona Chinensis Benth extracts" the authors studied the antioxidant and hypolipidemic effects of Mesona Chinensis Benth extracts. The article is well written and has a clear structure. The scheme of the experiment is understandable and sufficiently substantiated. The results obtained are well described and correctly interpreted. The presentation of the results is consistent and clear. Figures and tables are quite informative.

There are several small remarks to work:

1. The article would be more informative if the authors described the novelty of the scientific research in more detail.

2. Shortcomings in the English language, for example:

line 66: "of its hyperlipidemia 65 effec remains unclear".

Author Response

Dear reviewer:

    Thank you very much for your comments and suggestions.

Reply to your first suggestion: The novelty is the chemical and pharmacological evaluation of MCB extracts and seven fractions (line 74). Many parts of the article focus on chemical composition and biological activities (line 63, 382). It provided a theoretical basis for revealing the chemical components of MCB in biological activities.

Reply to your second suggestion: The article language was examined, including line 64: "of its hyperlipidemia 65 effective remains unclear". And the language of the article has been revised by a professional company.

Kind regards

Ping Yan